# AI Unsheathed: Testing Human–AI Collaboration Through Deck Construction in Competitive Strategy Games

## Abstract

This paper investigates the role of artificial intelligence as a collaborator in scientific reasoning by framing a known-answer question in the domain of competitive strategy games. Using Flesh and Blood (FaB) and the hero Kassai as a test case, the study evaluates whether AI can navigate both formal constraints and contextual judgment through four hypotheses: metagame identification, mainboard construction, card evaluation, and sideboard design. Results show that while the AI reproduces broad descriptive patterns and aligns well with community consensus, it systematically underestimates dominant strategies, misclassifies card roles, and fails to anticipate dynamic shifts in the competitive environment. Its outputs reflect biases inherited from online sources, revealing limitations in structural reasoning and contextual adaptation. Nonetheless, the AI demonstrates competence in organizing information, synthesizing consensus knowledge, and producing structured outputs. These findings highlight both the promise and the limits of human–AI collaboration: AI can serve as a valuable assistant in knowledge synthesis, but expert oversight remains indispensable to ensure accuracy, contextual awareness, and strategic depth.

*Keywords*: Experimental Economics, Artificial Intelligence, Strategy

# 1 Introduction: Human–AI Collaboration

This section introduces the intellectual framework for the study. It situates the role of AI within economics, a discipline well-suited for testing machine reasoning because of its emphasis on optimization, formal constraints, and reproducible logic. At the same time, it highlights the limitations of AI systems when applied to domains requiring contextual sensitivity, policy relevance, and creativity. By drawing on established theory and empirical findings, the discussion motivates the guided co-authorship protocol adopted in this paper. Economics can be viewed as the science of optimization in human society: individuals and firms choose under constraints, and their decentralized choices interact through prices to yield a competitive equilibrium that clears markets; under standard assumptions, such equilibria exist and connect to efficiency via the welfare theorems (Arrow and Debreu, 2024; Mas-Colell et al., 1995). In this setting, modern AI assists by scaling prediction and inference over high-dimensional data; empirical "scaling laws" further show systematic performance gains as data, model size, and compute increase, helping analysts interrogate richer environments at unprecedented scope (Athey, 2018; Kaplan et al., 2020). A central limitation, however, is whether AI can generate appropriate policy recommendations—i.e., sensitive to context, institutions, and strategic behavioral responses—because prediction alone may fail when policies change incentives. The Lucas critique cautions that models fit to historical correlations can become unreliable once rules shift, underscoring the need for causal structure and domain understanding (Lucas Jr, 1976). Likewise, evidence from social science emphasizes external-validity challenges: results (even from RCTs) need careful transport across settings (Deaton and Cartwright, 2018). Real deployments also reveal how objective misspecification can encode inequity, as in a widely used health-management algorithm that under-referred sicker Black patients because it optimized predicted cost rather than need, illustrating why expert oversight is indispensable (Obermeyer et al., 2019). Creativity research similarly indicates that, without human supervision, current systems struggle to reach expert-level performance. Studies find LLM outputs often appear prototypical or generic relative to professional writers, and broader analyses flag limits rooted in next-token prediction rather than autonomous, goal-directed agency (Ismayilzada et al., 2024; Bender et al., 2021). Conceptual work in computational creativity further argues that genuine creative advancement typically requires agency to transform conceptual spaces, a capacity machines lack absent human scaffolding (Ritchie, 2006). Taken together, available evidence suggests that AI creativity generally trails top-expert human performance and tends toward genericity unless guided (Ismayilzada et al., 2024). These limitations imply AI is unlikely to formulate truly pertinent research questions on its own. Accordingly, this paper adopts a guided co-authorship protocol: the human author supplies the ordered reasoning steps and domain constraints; the AI generates the prose and organizes citations; and the human verifies logical coherence and contextual appropriateness without otherwise altering the AI's text—preserving non-interference while maintaining expert oversight. Evidence on human–AI teaming supports this division of labor: optimizing for team, not standalone model, performance improves outcomes when humans calibrate reliance and provide contextual judgment (Bansal et al., 2019, 2021). This approach is also consistent with the conference's outcome-oriented evaluation—expert assessment of whether AI-generated papers make a contribution—and with the pragmatic view of AI as a partner that returns scarce time to scientists. Field and experimental studies find that generative-AI assistance can materially raise productivity and reduce task time in knowledge-work settings, especially for non-experts, supporting the claim that AI can shoulder execution while humans focus on validation and interpretation (Brynjolfsson et al., 2025; Noy and Zhang, 2023). Finally, the paper's goal is to evaluate whether the AI can provide appropriate answers to a concrete social-science problem for which accepted solutions exist and supporting data are publicly available. The study leverages the growing infrastructure for transparency and open data in the social sciences, which facilitates independent verification and reproducibility of claims (Christensen and Miguel, 2018; Gentzkow and Shapiro, 2014).

# 2 Methodology: Asking a Known-Answer Question

This section outlines the methodological approach of the study. It explains why the experiment is framed as a known-answer question, how this design follows the broader goals of the Agents4Science conference, and why expert evaluation is necessary. It then introduces the choice of Trading Card Games (TCGs) as a test domain, showing how their combination of formal rules, optimization, and social complexity provides a rigorous yet accessible setting to evaluate whether AI can reason and demonstrate contextual judgment in a manner comparable to human researchers. The Agents4Science

conference, organized by Stanford University, explicitly frames its purpose as an open investigation into whether AI can produce useful scientific papers. As the committee acknowledges, "we don't know yet, and that is exactly why this conference exists" (Agents4Science Committee, 2025). Rather than presuming AI's effectiveness as a collaborator, the conference operates as a transparent experiment in which AI agents serve as both authors and reviewers, with their outputs subjected to the same scrutiny as human work. This design reflects the committee's commitment to empirical assessment, treating both successes and failures as informative for understanding AI's evolving role in science. Because no prior conclusions exist, the conference necessarily requires a methodology for evaluation. This entails delegating review of AI-generated manuscripts to domain experts, who can identify logical inconsistencies, methodological flaws, and contextual oversights that automated systems are unlikely to detect. Such expert involvement safeguards academic standards while creating a structured framework for systematic critique, ensuring that the experiment yields evidence on both the potential and the limitations of AI. At the core of this experiment lies a crucial question: can AI demonstrate the contextual judgment that human scientists instinctively recognize as correct? Scientific writing extends beyond logical inference; it demands sensitivity to disciplinary norms, relevance of evidence, and appropriateness of interpretation. Reviewers will therefore judge AI-generated work primarily on its ability to display such judgment. Without it, even technically accurate writing risks being dismissed as superficial, underscoring why contextual reasoning remains the decisive benchmark for AI's role in research. To probe this, the present study adopts a design in which the human author poses a question with a known answer, enabling independent verification of the AI's response. The problem is selected to emulate the process of economic science, where most arguments can be evaluated through logical coherence and empirical consistency, while a minority requires subjective reviewer judgment. This mirrors academic practice: claims are largely verifiable, yet interpretation and contextualization demand expert discretion. As a testbed, one possibility is to require the AI to construct a specific trading card game (TCG) strategy. Such a task combines the rigor of constrained optimization under explicit rules—akin to economic modeling—with the evaluative component of assessing contextual plausibility and creativity. TCGs are especially suited to this purpose. Popular among IT professionals and engineers, they resemble complex mathematical puzzles that require managing constraints, optimizing resources, and anticipating adversarial play. Competitive players treat the activity as a strategic sport aimed at exploiting marginal advantages. Yet participation is costly: a single competitive strategy often requires around USD 500, excluding international travel, restricting the scene largely to high-income professionals. This exclusivity is offset by substantial tournament prize pools, often worth several thousands of dollars, which foster an intensely competitive environment and drive the development of highly optimized strategies. Beyond their mathematical rigor, TCGs also carry a social science dimension. While governed by deterministic, reproducible rules, they exist as socially embedded practices supported by abundant online discourse—guides, forums, and streams—that AI can readily access. However, strategies rarely undergo rigorous analysis; instead, players adopt personal versions shaped by preferences, local metagames, and peer influence, often without justification. This abundance of inconsistent or low-quality content risks misleading AI systems. Moreover, the publisher itself periodically alters the environment by releasing new cards and mechanics, ensuring that strategies remain fluid rather than definitive. TCGs thus present a dual challenge for AI: reasoning within fixed formal rules while adapting to a dynamic, socially constructed context.

## 2.1 Experiment and Hypothesis

The experiment asks the AI to generate a competitive decklist for the hero Kassai from Flesh and Blood (FaB). FaB is chosen because it is widely regarded as the most complex commercial card game, requiring players to navigate intricate rules, deep resource management, and constant adaptation to opponents. Mastering a single strategy typically demands hundreds of games and several months of practice, reflecting both the intellectual depth of the game and the substantial effort required to achieve competitive proficiency. Kassai is particularly well suited for this study for three reasons. First, she is an interactive hero, requiring players to adapt continuously to their opponents' actions, which makes strategic reasoning central to her gameplay. Second, her deck construction is complex, since many card inclusions are debatable and can shift depending on context, offering a rich ground for testing evaluative choices. Third, she represents a developing strategy: Kassai is not considered one of the strongest heroes, but she is regularly improved through new card releases, gradually increasing her competitiveness and keeping her strategic environment in flux. This study does not employ experimental treatments. Instead, the methodology is structured as a progressive deck

construction process, carried out in four steps. Each step mirrors one of the four hypotheses outlined in the paper, allowing the experiment to evaluate AI performance in a systematic and cumulative way. By moving from metagame identification to sideboard design, the approach ensures that every stage of deckbuilding is explicitly tied to a testable claim about AI's reasoning and judgment:

**Hypothesis 1:** AI can accurately identify the composition of the metagame—that is, the dominant strategies and their relative frequencies expected to be played by other participants in the tournament.

**Hypothesis 2:** AI can accurately construct the mainboard for the metagame—namely, the 60 cards forming the core of the strategy, selected for their efficiency with the chosen hero in the given environment.

**Hypothesis 3:** AI can assess the importance of individual cards within a strategy by classifying them into four categories: (i) Power cards, Core elements that define the strategy and directly drive victory (ii) Staples, Highly efficient, widely used cards that provide consistency across strategies (iii) Support cards, Tools that enable or enhance the main plan, often by countering opponents or smoothing resource use (iv) Fillers, Marginal cards included mainly to complete the deck, offering limited impact but ensuring the required card count.

**Hypothesis 4:** AI can accurately construct the sideboard for the metagame—that is, the additional 15 cards designed to respond to atypical or situational strategies encountered in tournament play.

These hypotheses together provide a comprehensive panorama of what can be achieved with the strategy, with the exception of the precise configurations to play in each matchup. Those configurations are examined under Hypothesis 3, where the AI is asked to justify which choices apply to which matchups.

## 3 Results

The results demonstrate that ChatGPT cannot independently interpret the table correctly; its estimations remain descriptive outputs that lack contextual awareness. As a result, the AI must be guided in its interpretation by the human author, who provides the necessary domain expertise and methodological framing to distinguish between open-entry and selective-entry dynamics, assess the impact of the BnR, and evaluate the plausibility of the strategy distributions.

### 3.1 Hypothesis 1

Table 1 reports the AI's estimation of the metagame on September 3, 2025, just after the Banishment and Restricted (BnR) list of September 1 that concluded the High Seas competitive season, providing a fully resolved dataset; these estimates are compared to the Pro Tour Singapore and Week 1 benchmarks. The results show that the AI systematically underestimates dominant strategies such as Arakni S, Gravy Bones, Cindra, and Verdance, likely because its perspective mirrors open-entry tournaments where average players follow personal preferences under financial limits, rather than selective-entry events where elite competitors focus exclusively on the strongest decks. By contrast, the AI overestimates Fang, a secondary but constrained strategy more common in open-entry play, while underestimating its more versatile variant Kassai, which is favored at higher levels of competition. It further aggregates tertiary strategies instead of evaluating them individually, acknowledging their minor but highly contextual presence. Finally, comparison with Week 1 data reveals that the AI does not incorporate the evolution of power dynamics following the BnR, continuing to overvalue weakened top decks while undervaluing secondary strategies that gained from the changes, in contrast to human competitors who naturally adjust their expectations, albeit with variation across playgroups. In sum, Table 1 shows that while the AI efficiently estimates a standard competitive metagame, it fails to capture the adaptive strategies of top-level play and cannot anticipate future shifts in the metagame due to its lack of structural reasoning.

Table 1: Hypothesis 1

| Hero | Est. 1 | Est. 2 | Est. 3 | Est. 4 | Est. 5 | Est. 6 | Est. 7 | Est. 8 | Est. 9 | Est. 10 | Average | PT Sin | Week 1 |
|---|---|---|---|---|---|---|---|---|---|---|---|---|---|
| Arakni (S) | 20.8% | 9% | 20% | 5% | 6% | 18% | 18% | 12% | 12% | 21% | 14.18% | 21.5% | 5.10% |
| Others | 22% | 0% | 4% | 11% | 15% | 4% | 10% | 14% | 17.5% | 16% | 11.35% | – | – |
| Gravy Bones | 12.2% | – | 12% | 8% | 7% | 16% | 14% | 10% | 9% | 12% | 10.12% | 12.8% | 6.12% |
| Cindra | 11.9% | – | 11% | 9% | 4% | 7% | 10% | 9% | 4.5% | 12% | 7.84% | 12.5% | 7.76% |
| Dash I/O | 5.7% | 12% | 5% | 5% | 5% | 5% | 5% | 6% | 5.5% | 6% | 6.02% | 6% | 7.96% |
| Uzuri | – | 6% | – | – | – | – | – | – | – | – | 0.6% | – | % |
| Oscilio | 3.4% | – | 5% | – | 5% | 8% | 5% | 8% | 7.0% | – | 4.14% | 3.5% | 7.96% |
| Verdance | 6.8% | – | 6% | 6% | 4% | 5% | 7% | 4% | 3.5% | 7% | 4.93% | 7.1% | 5.71% |
| Valda | – | – | – | – | 7% | 6% | 3% | 5% | 8.0% | 3% | 3.2% | – | 1.63% |
| Prism | 5.5% | 11% | 5% | 8% | 5% | 2% | 3.5% | 4% | 2.5% | 5% | 5.15% | 5.7% | 4.90% |
| Ira | 3.9% | – | 3% | – | 5% | 6% | 5% | 6% | 6.0% | – | 3.49% | 3.3% | 4.29% |
| Florian | – | – | – | 12% | – | – | – | 4% | 1.0% | 2% | 1.9% | 2.4% | 6.53% |
| Fang | – | – | 3% | 4% | 4% | 4% | 4% | 5% | 5.0% | – | 2.9% | 1.4% | 5.31% |
| Vynnset | – | 6% | – | 6% | 4% | 1.5% | – | 1.0% | – | – | 1.85% | 2.4% | 4.08% |
| Kano | 2.3% | 6% | 4% | 3% | 4% | 2% | 3% | 3% | 3.0% | 2% | 3.23% | 4.1% | 2.04% |
| Bravo | – | 5% | 1% | – | – | – | – | – | – | – | 0.6% | 0.3% | – |
| Katsu | – | 7% | 2% | – | 2% | 2% | 1% | – | 1.5% | 3% | 1.85% | 1.6% | 2.45% |
| Arakni (M) | – | – | 3% | 1% | 1% | 3% | 1.5% | 4% | 5.0% | 2% | 2.05% | 2.4% | % |
| Kassai | 2.5% | – | 3% | 4% | 2% | – | 1.5% | 3% | 1.5% | – | 1.75% | 2.7% | 2.04% |
| Victor | 2.5% | – | 3% | – | 4% | 1% | 2% | 3% | 1.5% | 3% | 2% | 4.1% | 2.65% |
| Fai | – | 4% | – | – | – | 2% | – | – | 2% | – | 0.8% | – | 0.20% |
| Rhinar | – | 4% | 2% | – | 3% | 1% | 0.8% | – | – | – | 1.08% | 1.1% | 1.63% |
| Boltyn | – | 2% | – | – | – | – | – | – | – | – | 0.2% | – | % |
| Dorinthea | – | 4% | 2% | – | 3% | 1% | 0.8% | – | 1.5% | 1% | 1.33% | 1.4% | 2.04% |
| Riptide | – | 2% | 2% | – | 3% | 1.5% | 1% | – | – | – | 0.95% | 1.6% | 1.84% |
| Jarl | – | – | 1% | – | 2% | 2% | 1% | – | 1.5% | 3% | 1.05% | 1.6% | 2.45% |
| Levia | – | 3% | 1% | – | 2% | 1% | 0.5% | – | – | – | 0.75% | 0.3% | 1.02% |
| Kayo | – | – | 1% | – | – | – | – | – | – | – | 0.1% | 1.1% | 3.88% |
| Marlynn | – | – | 1% | – | – | – | – | – | – | – | 0.1% | 0.8% | 1.22% |
| Maxx Nitro | – | – | 1% | – | 1% | – | – | – | – | – | 0.1% | 0.3% | 0.20% |
| Puffin | – | – | 1% | – | 1% | – | – | – | – | – | 0.1% | 0.3% | 3.47% |
| Teklovossen | – | – | 1% | – | – | 1% | – | – | – | – | 0.1% | 0.3% | – |
| Arakni (H) | – | – | – | – | – | 1% | – | – | – | – | – | – | 0.61% |
| **Not legal** | | | | | | | | | | | | | |
| Azalea | – | 5% | – | 7% | – | – | 2% | – | – | – | 1.4% | – | – |
| Dromai | – | 8% | – | – | – | – | – | – | – | – | 0.8% | – | – |
| Iyslander | – | 6% | – | – | – | – | – | – | – | – | 0.6% | – | – |
| Dash | – | – | – | – | – | – | – | – | – | 2% | 0.2% | – | – |
| Aurora | – | – | – | – | 1% | – | – | – | – | – | 0.1% | – | – |
| **Total** | 99.5% | 100% | 103% | 89% | 100% | 100% | 99.6% | 100% | 100% | 100% | – | – | – |
| Observations | – | – | – | – | – | – | – | – | – | – | – | 380 | 445 |
| Tier 4 (Day 1) | Yes | Yes | Yes | Yes | Yes | Yes | Yes | Yes | Yes | Yes | – | – | – |
| Tier 4 (Day 2) | Yes | No | Yes | Yes | No | No | Yes | No | No | Yes | – | – | – |
| Tier 3 | No | Yes | Yes | Yes | Yes | Yes | Yes | Yes | Yes | No | – | – | – |
| Tier 2 | No | No | No | No | No | No | No | No | No | No | – | – | – |
| Online data | No | No | Yes | No | Yes | Yes | Yes | Yes | No | No | – | – | – |
| Online articles | No | Yes | No | Yes | No | No | No | No | No | No | – | – | – |
| Cards legality | No | Yes | No | Yes | No | Yes | No | Yes | Yes | Yes | – | – | – |

## 3.2 Hypothesis 2, 3 & 4

The results of Hypothesis 2 show that the AI is generally accurate at estimating the mainboard composition, but its accuracy depends heavily on human consensus. The average quantity (AvgQ) column indicates that the AI makes correct estimations for cards that are consistently present across reference decklists, reflecting strong community agreement. However, for cards where inclusion varies with player interpretation (TarQ), AvgQ should instead be read as an indicator of the probability that a card appears in the deck rather than as a strict prescription. This means the AI mirrors consensus well, but also reproduces human errors and biases. For example, it assumes Hit and Run (Red) and Gorganian Tome are automatic inclusions despite being mediocre, overvalues Hit and Run (Yellow) while neglecting Outland Skirmish (Yellow), and overweights Draw Swords (Blue) compared to Overpower (Blue). It also fails to recognize that Slice and Dice (Red) is always played in three copies, and that Rise an Army is a sideboard card never played mainboard. Overall, the AI reflects human reasoning from a limited sample: it is accurate when consensus is accurate, but biased when consensus is flawed, showing that it reproduces descriptive patterns without deeper structural understanding.

The results of Hypothesis 3 indicate that while the AI makes broadly correct estimations of card quality, its evaluations are shaped by biases it inherits from human-written sources. The average rating (AvgR) column shows that the AI's qualitative judgments are often influenced by textual analysis drawn from online articles rather than from structural, competitive reasoning. This leads to

Table 2: Hypothesis 1

| Card | L1 | L2 | L3 | L4 | L5 | L6 | L7 | L8 | L9 | L10 | AvgQ | AvgR | Mode | TarQ | TarR |
|---|---|---|---|---|---|---|---|---|---|---|---|---|---|---|---|
| **Equipment** | | | | | | | | | | | | | | | |
| Cintari Saber | - | 2(S) | 2(P) | 2(S) | 2(S) | 2(S) | 2(P) | 2(P) | 2(S) | 2(S) | 2 | 3.3 | 2(S) | 2(S) | 3 |
| Crown of Dominion | 1(S) | 1(U) | 1(S) | 1(S) | 1(U) | 1(S) | 1(S) | 1(S) | 1(S) | 1(U) | 1 | 2.7 | 1(S) | 1(S) | 2.8 |
| Balance of Justice | - | - | - | - | - | - | - | - | - | 1(U) | 0.1 | 2 | 1(U) | - | - |
| Braveforge Bracers | 1(S) | 1(P) | 1(S) | 1(S) | 1(S) | 1(S) | 1(S) | 1(S) | 1(S) | 1(S) | 1 | 3.1 | 1(S) | 1(U) | 2.6 |
| Hot Streak | - | - | - | - | - | - | - | - | 1(U) | - | 0.1 | 2 | 1(U) | - | - |
| Valiant Dynamo | 1(S) | 1(S) | 1(S) | 1(S) | 1(U) | 1(S) | 1(S) | 1(S) | 1(S) | 1(S) | 1 | 2.9 | 1(S) | 1(P) | 3.2 |
| Grains of Bloodspill | 1(U) | 1(U) | 1(U) | 1(U) | 1(U) | 1(S) | 1(P) | 1(U) | - | - | 0.8 | 2.38 | 1(U) | 1(P) | 3.4 |
| Nullrune Gloves | 1(U) | - | - | - | - | - | - | - | 1(U) | - | 0.2 | 2 | 1(U) | - | - |
| Nullrune Boots | 1(U) | - | - | - | - | - | - | - | - | - | 0.1 | 2 | 1(U) | - | - |
| **Mainboard** | | | | | | | | | | | | | | | |
| Red | | | | | | | | | | | | | | | |
| Blade Flurry | 3(S) | 3(P) | 3(P) | 3(P) | - | 3(P) | 3(P) | 3(S) | 3(P) | 3(P) | 2.7 | 3.78 | 3(P) | 3(P) | 3.2 |
| Blanch | - | - | - | - | - | 3(U) | - | - | - | - | 0.3 | 2 | 3(U) | - | - |
| Blade Runner | 3(S) | - | - | - | - | - | - | 3(S) | - | 3(S) | 0.9 | 3 | 3(S) | 3(S) | 3 |
| Command and Conquer | - | - | - | - | 2(P) | - | - | - | 2(P) | - | 0.4 | 4 | 2(P) | - | - |
| Draw Swords | 3(U) | 3(P) | 3(P) | 3(P) | 3(P) | 3(P) | 3(P) | 3(P) | 2(P) | 3(S) | 2.9 | 3.7 | 3(P) | 3(S) | 3 |
| Fate Foreseen | - | - | 3(U) | 2(U) | 2(U) | 1(S) | 1(S) | - | - | - | 0.6 | 2.5 | 2(U)/1(S) | - | - |
| Hit and Run | 3(S) | 2(S) | - | - | 3(S) | 3(S) | 3(S) | 3(S) | 3(S) | - | 2 | 3 | 3(S) | - | - |
| In the Swing | 3(S) | 3(S) | 3(S) | 3(S) | 2(U) | 3(P) | 3(S) | 3(S) | 3(P) | 3(S) | 2.9 | 3.1 | 3(S) | 3(S) | 3 |
| Ironsong Response | - | 2(U) | - | - | - | 2(U) | 2(U) | - | - | - | 0.6 | 2 | 2(U) | - | - |
| Nourishing Emptiness | - | - | - | - | 1(P) | 1(P) | 1(P) | - | - | 2(P) | 0.5 | 4 | 1(P) | - | - |
| Outland Skirmish | 3(S) | 3(S) | 3(S) | 3(S) | 3(S) | 3(S) | 3(S) | 3(S) | 3(S) | 3(S) | 3 | 3 | 3(S) | 3(U) | 2.6 |
| Performance Bonus | - | - | - | - | - | 1(U) | 1(U) | - | - | - | 0.2 | 2 | - | - | - |
| Sharpened Senses* | - | 3(P) | - | - | - | - | - | - | - | - | 0.3 | 4 | 3(P) | - | - |
| Shelter from the Storm | 1(U) | - | 1(U) | - | 2(U) | 2(S) | 2(U) | - | - | - | 0.8 | 2.2 | 1(U)/2(U) | - | - |
| Slice and Dice | 3(P) | 3(P) | 3(P) | 3(P) | 2(U) | - | 3(P) | 3(S) | 1(U) | 2(S) | 2.3 | 3.33 | 3(P) | 3(S) | 3 |
| Spoils of War | 3(P) | 3(P/S) | 3(P) | 3(P) | 3(P) | 3(P) | 3(P) | 3(P) | 3(P) | 3(P) | 3 | 3.95 | 3(P) | 3(P) | 4 |
| Unsheathed | 3(U) | 3(P) | 3(S) | 3(S) | 3(P) | 3(P) | 2(U) | 3(P) | 2(S) | 3(P) | 2.8 | 3.3 | 3(P) | 3(P) | 3.8 |
| Take it on the Chin | - | - | - | - | - | 2(U) | 2(U) | - | - | - | 0.4 | 2 | 2(U) | - | - |
| Yellow | | | | | | | | | | | | | | | |
| Blade Runner | 3(S) | - | - | - | - | 1(U) | 1(U) | 3(S) | 2(S) | 3(S) | 1.3 | 2.67 | 3(S) | 3(U) | 2.6 |
| Blood on Her Hands | 3(P) | 3(P) | 3(P) | 3(P) | 3(P) | 3(P) | 3(P) | 3(P) | 3(P) | 3(P) | 3 | 4 | 3(P) | 3(P) | 3.6 |
| Draw Swords | - | 3(S) | 3(P) | 3(P) | 2(U) | 1(U) | 3(S) | 3(S) | 2(S) | 2(U) | 2.2 | 2.89 | 3(S) | 3(F) | 2.4 |
| Hit and Run | 3(S) | 3(S) | 3(S) | 3(S) | 3(S) | 3(S) | - | 3(S) | 3(S) | - | 2.4 | 3 | 3(S) | | |
| Outland Skirmish | - | - | - | - | - | - | - | - | - | - | - | - | - | 3(F) | 2.2 |
| Raise an Army | 3(U) | 2(U) | 3(U) | 3(U) | 2(U) | 2(U) | 2(U) | - | 3(P) | 3(U) | 2.3 | 2.22 | 2(U)/3(U) | - | - |
| Riches of Tropal Dhani | - | - | - | - | - | - | - | - | 1(U) | - | 0.1 | 2 | 1(U) | 1(S) | 3 |
| Run Through | 3(S) | 3(S) | 3(S) | 3(S) | 3(S) | 3(S) | 3(S) | 1(U) | 2(S) | 3(S) | 2.7 | 2.9 | 3(S) | 3(U) | 2.6 |
| Sharpened Senses | 3(S) | 3(P) | 3(S) | 3(S) | 3(P) | 3(S) | 3(S) | 3(S) | 3(P) | - | 2.7 | 3.33 | 3(S) | 3(U) | 2.6 |
| Slice and Dice | - | 2(U) | - | 2(S) | - | - | - | 2(S) | - | - | 0.6 | 2.67 | 2(S) | 3(S) | 2.8 |
| That All You Got ? | - | - | - | - | 2(U) | 2(U) | - | - | 2(U) | 2(U) | 0.8 | 2 | 2(U) | - | - |
| Blue | | | | | | | | | | | | | | | |
| Amulet of Echoes | - | - | - | - | - | - | - | - | 1(F) | - | 0.1 | 1 | 1(F) | - | - |
| Blade Flurry* | - | - | - | - | 2(U) | - | - | - | - | - | 0.2 | 2 | 2(U) | - | - |
| Blade Runner | 3(U) | 3(S) | 3(S) | 3(S) | - | - | 2(S) | 3(S) | 3(S) | 3(S) | 2.3 | 2.88 | 3(S) | 3(F) | 2 |
| Draw Swords | - | 2(S) | 2(S) | 2(U) | - | - | - | - | - | - | 0.6 | 2.67 | 2(S) | - | - |
| Eye of Ophidia | 1(U) | 1(U) | 1(U) | - | - | - | 1(U) | - | 1(U) | - | 0.5 | 2 | 1(U) | - | - |
| Glint the Quicksilver | 3(S) | 3(P/S) | 3(S) | 3(P) | 3(S) | 3(S) | 3(S) | 3(S) | 3(S) | 3(P) | 3 | 3.25 | 3(S) | 3(S) | 3 |
| Hit and Run | 3(U) | 3(S) | 3(S) | 3(S) | 3(S) | 2(S) | 3(S) | 3(S) | 3(S) | 3(S) | 2.9 | 2.9 | 3(S) | 3(U) | 2.8 |
| Outland Skirmish | - | - | - | - | - | - | 2(F) | - | - | - | 0.2 | 1 | 2(F) | - | - |
| Overpower | - | - | - | - | - | - | - | 1(F) | - | - | 0.1 | 1 | 1(F) | 3(U) | 2.6 |
| Precision Press | - | - | 2(U) | - | - | - | 2(U) | - | - | 3(U) | 0.7 | 2 | 2(U) | - | - |
| Provoke | 1(F) | - | - | 3(U) | 3(U) | 2(U) | 2(U) | 3(U) | 3(U) | 3(U) | 2 | 1.88 | 3(U) | - | - |
| Snag | - | - | 2(U) | - | - | - | - | - | - | - | 0.2 | 2 | 2(U) | - | - |
| Trot Along | 3(U) | 2(U) | 3(U) | 3(U) | 3(U) | 2(U) | 3(U) | 3(U) | 3(U) | 2(U) | 2.7 | 2 | 3(U) | 3(U) | 2.8 |
| Gorganian Tome* | 1(U) | 1(P) | 1(P) | 1(U) | 1(U) | - | 1(U) | 1(U) | - | 1(U) | 0.9 | 2.5 | 1(U) | - | - |
| **Sideboard** | | | | | | | | | | | | | | | |
| Red | | | | | | | | | | | | | | | |
| Battlefront Bastion | | | | | | | | | | | | | | 3(U) | 2.4 |
| Blanch | - | - | - | - | - | - | - | - | 2(U) | - | 0.2 | 2 | 2(U) | - | - |
| Command and Conquer | 2(S) | 2(S) | - | - | 1(P) | 2(P) | 2(S) | 2(S) | - | 2(P) | 1.3 | 3.43 | 2(S) | - | - |
| Fate Foreseen | 2(U) | 2(U) | 3(U) | - | - | - | 2(U) | 3(U) | 1(U) | 2(S) | 1.5 | 2.14 | 2(U) | 3(S) | 3 |
| Ironsong Response | - | - | 2(S) | - | - | - | - | - | - | - | 0.2 | 3 | 2(S) | - | - |
| Kabuto of Imperial Authority | | | | | | | | | | | | | | 1(U) | 2.4 |
| Nourishing Emptiness | 1(P) | 1(U/F) | - | 3(P) | 1(P) | - | - | 1(P) | 1(P) | - | 0.8 | 3.58 | 1(P) | - | - |
| Nullrune Boots | | | | | | | | | | | | | | 1(U) | 2.4 |
| Nullrune Gloves | | | | | | | | | | | | | | 1(U) | 2.4 |
| Performance Bonus | - | - | - | 1(U) | 1(U) | - | 1(U) | - | - | - | 0.3 | 2 | 1(U) | - | - |
| Take it on the Chin | - | - | - | - | 2(U) | - | 2(U) | - | - | - | 0.4 | 2 | 2(U) | - | - |
| Shelter from the Storm | 1(U) | 2(U) | 1(U) | 2(U) | - | 1(S) | 3(U) | 2(U) | 2(U) | 2(U) | 1.6 | 2.11 | 2(U) | 3(U) | 2.4 |
| Sink Below | - | - | - | - | - | - | - | 1(U) | 2(S) | - | 0.3 | 2.5 | 1(U)/2(S) | - | - |
| Slice and Dice | - | - | - | 1(U) | - | - | - | 2(U) | - | - | 0.3 | 2 | 1(U)/2(U) | - | - |
| Stroke of Foresight | - | - | - | 2(S) | - | - | - | - | - | - | 0.2 | 3 | 2(S) | - | - |
| Yellow | | | | | | | | | | | | | | | |
| Cash In | - | - | 2(U) | - | - | - | - | - | - | - | 0.2 | 2 | 2(U) | - | - |
| Raise an Army | - | - | - | - | 1(U) | 1(U) | - | 2(U) | - | - | 0.4 | 2 | 1(U) | - | - |
| Riches of Tropal Dhani | - | - | 1(U) | - | - | - | - | - | 1(U) | - | 0.2 | 2 | 1(U) | - | - |
| That All You Got ? | 2(U) | 2(U) | 2(U) | 2(U) | 1(U) | 1(U) | 2(U) | 1(U) | - | - | 1.3 | 2 | 2(U) | - | - |
| Seduce Secrets | 2(U) | - | - | - | 2(U) | - | - | - | - | - | 0.4 | 2 | 2(U) | - | - |
| Slice and Dice | - | - | - | - | - | 2(S) | - | - | - | - | 0.2 | 3 | 2(S) | - | - |
| Blue | | | | | | | | | | | | | | | |
| Blade Runner | - | - | - | - | - | 1(U) | - | - | - | - | 0.1 | 2(U) | 1(U) | - | - |
| Draw Swords | - | - | - | - | 2(U) | 1(U) | - | - | - | - | 0.3 | 2 | 1(U)/2(U) | - | - |
| Eye of Ophidia | - | - | 1(U) | - | - | - | - | - | - | - | 0.1 | 2 | 1(U) | - | - |
| Gorganian Tome | - | - | - | - | - | 1(U) | - | - | 1(U) | - | 0.2 | 2 | 1(U) | - | - |
| Provoke | 2(F) | 2(U) | - | - | - | - | - | - | - | - | 0.4 | 1.5 | 2(F)/2(U) | - | - |
| This Round's on Me | - | - | 2(U) | - | - | 1(U) | - | - | - | 2(U) | 0.5 | 2 | 2(U) | - | - |
| Slice and Dice | - | - | - | - | - | 2(S) | - | - | - | - | 0.2 | 3 | 2(S) | - | - |
| Snag | 2(U) | 3(U) | - | 2(U) | 2(U) | - | 2(U) | 2(U) | 2(U) | - | 1.5 | 2 | 2(U) | - | - |
| Steelblade Shunt | - | - | - | - | - | - | - | - | 2(U) | - | 0.2 | 2 | 2(U) | - | - |
| Total | 81 | 82 | 79 | 80 | 79 | 79 | 86 | 80 | 80 | 78 | - | - | - | - | - |

several notable misclassifications from the player's understanding (TarR). For example, it labels *Braveforge Bracers* as a Staple instead of a standard card, treats *Valiant Dynamo* as a Staple without recognizing it as a true power card, and misjudges *Grains of Bloodspill* as a Support card rather than a power card. These errors reveal a superficial perspective more typical of content creators than expert players, focusing on appearance rather than functional dynamics. Similar issues emerge in the evaluation of mainboard cards: *Blade Flurry* and *Draw Swords* are rated as Power cards, when in reality they function as Staples that only approach Power status under specific conditions; *Blood on Her Hands* is considered a top Power card without acknowledging its constraints; and *Unsheathed* is undervalued as a Staple instead of being recognized as the second-best card in the deck. Conversely, weaker or situational cards such as *Enhanced Senses* and *Blade Runner* (Blue) are overrated as Staples, while context-dependent but strategically strong supports like *Trot Along* and *Riches of Tropal Dhani* are downgraded to Fillers, alongside *Overpower*, which is misclassified due to its niche utility. Taken together, these results suggest that the AI captures the general power level of cards but consistently falls prey to human biases, producing evaluations that are directionally correct yet flawed whenever deeper structural understanding is required.

The results of Hypothesis 4 demonstrate that the AI does not understand how to build an effective sideboard, as it lacks the contextual judgment required for proper deck construction. A well-designed sideboard should address Warrior's typical weaknesses: against Illusionist with Battlefront Bastion, against Wizard with Nullrune Boots and Nullrune Gloves, and in the mirror matchup with Kabuto of Imperial Authority to avoid an automatic loss. The AI fails to identify any of these cards as relevant. Its only partially correct decision is recognizing that red Defense Reactions such as Fate Foreseen and Shelter from the Storm belong in the sideboard. However, it misinterprets their purpose, failing to include the necessary three copies of each to enable a strong defensive configuration when the deck must play reactively. This misstep illustrates the AI's broader pattern: it treats the sideboard as a place for sprinkling one or two situational cards, without grasping their strategic function. The unnecessary inclusion of Yellow and Blue cards that do not serve as sideboard material further highlights the lack of structural awareness. Instead of applying the deeper logic of sideboard construction, the AI merely reproduces superficial patterns, showing it cannot translate card knowledge into functional deckbuilding decisions.

# 4    Conclusion

This study examined whether AI could contribute to competitive deck construction in Flesh and Blood, organized around four hypotheses. The results show that while AI can approximate human reasoning, its limits become clear when deeper structural insight is required. For Hypothesis 1, the AI generated a plausible but flawed description of the metagame: it systematically underestimated dominant strategies, overestimated weaker ones, and failed to anticipate post-ban adjustments, reflecting descriptive replication rather than adaptive foresight. For Hypothesis 2, the AI successfully mirrored community consensus in assembling the mainboard, but it also reproduced human biases and errors, misvaluing certain inclusions and failing to distinguish between sideboard and core cards. Hypothesis 3 revealed that its card evaluations were broadly correct in direction but shaped by biases inherited from online sources, leading to misclassifications of both powerful and situational cards. Finally, Hypothesis 4 showed that while the AI could propose reasonable sideboard options, its reasoning remained superficial, overlooking the contextual nuances that guide expert deck adjustments. Taken together, these findings highlight the dual nature of human–AI collaboration: the AI is adept at synthesizing consensus knowledge and producing structured outputs, yet expert oversight remains indispensable for ensuring accuracy, contextual awareness, and strategic depth.

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

# A  Agents4Science AI Involvement Checklist

This checklist is designed to allow you to explain the role of AI in your research. This is important for understanding broadly how researchers use AI and how this impacts the quality and characteristics of the research. **Do not remove the checklist! Papers not including the checklist will be desk rejected.** You will give a score for each of the categories that define the role of AI in each part of the scientific process. The scores are as follows:

- **[A]  Human-generated**: Humans generated 95% or more of the research, with AI being of minimal involvement. 0/10
- **[B]  Mostly human, assisted by AI**: The research was a collaboration between humans and AI models, but humans produced the majority (>50%) of the research. 3/10
- **[C]  Mostly AI, assisted by human**: The research task was a collaboration between humans and AI models, but AI produced the majority (>50%) of the research. 7/10
- **[D]  AI-generated**: AI performed over 95% of the research. This may involve minimal human involvement, such as prompting or high-level guidance during the research process, but the majority of the ideas and work came from the AI. 0/10

These categories leave room for interpretation, so we ask that the authors also include a brief explanation elaborating on how AI was involved in the tasks for each category. Please keep your explanation to less than 150 words.

1. **Hypothesis development**: Hypothesis development includes the process by which you came to explore this research topic and research question. This can involve the background research performed by either researchers or by AI. This can also involve whether the idea was proposed by researchers or by AI. Answer: Inspired by my personal life and question I was wondering. Explanation: I know the answer to the question and want to know whether the AI can indicate it closely, or cannot grasp the contextual judgments.

2. **Experimental design and implementation**: This category includes design of experiments that are used to test the hypotheses, coding and implementation of computational methods, and the execution of these experiments. Answer: I ask the AI to generate 10 decklists. Explanation: It will be enough to judge whether it can provide accurate or approximate answer, given its tendency to focus on the same answers and my expertise on the topic.

3. **Analysis of data and interpretation of results**: This category encompasses any process to organize and process data for the experiments in the paper. It also includes interpretations of the results of the study. Answer: Date are Table 1 and Table 2. Explanation: The goal of the experiment is to have the AI generating the data and myself analyzing their correctness.

4. **Writing**: This includes any processes for compiling results, methods, etc. into the final paper form. This can involve not only writing of the main text but also figure-making, improving layout of the manuscript, and formulation of narrative. Answer: Chat GPT from my precise indications. Explanation: The AI cannot make correct writing if left alone. I let him liberties with the Literature Review only, to know which paper it was going to cite. Can provide the prompt used on demand.

5. **Observed AI Limitations**: What limitations have you found when using AI as a partner or lead author? Description: Must be told what to do precisely. Otherwise it is too vague.


# B   Prompts used to write the paper

## B.1   Introduction

Write the Introduction section of a research paper by following the specific steps of reasoning that I will provide. The reasoning steps must appear in the exact order in which I present them. Each step should be supported by evidence from published research, with citations drawn from Google Scholar. Citations should be inserted at the end of the relevant sentences. The final text should not exceed one page in Overleaf.

Economics is the science of optimization in human society. Explain what this entails and the process that leads to equilibrium. Explain how AI can assist in this task, since its computational power allows it to analyze data and draw conclusions at an unprecedented scale.

Explain why the limitation lies in whether AI can provide policy recommendations that are "appropriate," as this requires understanding the complexity and particularities of human interactions. Clarify that a lack of such understanding means AI will always require expert human oversight.

Explain why the creativity literature highlights the inability of AI to achieve expert-level performance without human supervision. AI creativity is generally inferior to that of the top 10% of humans and tends to produce generic answers. This indicates that AI cannot fully grasp advanced human considerations and cannot develop them due to its lack of agency.

Explain how this limitation implies that AI is unlikely to generate truly pertinent research questions, thereby obligating the human co-author to write this paper in partnership with the AI by guiding its reasoning. The original rules of the conference will be respected, with the author providing only the reasoning steps to the AI and not interfering with the final text, except to verify its logical coherence.

Explain how this approach and consistent with the implicit outcome of this conference, meaning evaluating the pertinency of the AI-generated papers by experts to determine whether they make a contribution. The human author anticipate on that the conclusion will be validated as a partner because it provides what scientist are looking for: time.

Explain how the goal of the paper will be the human author evaluating whether the AI can provide appropriate answers to a Social Science problem, since the more correct solutions of this problem are known and the data supporting these conclusions are available on Internet.

*A posteriori:* Please introduce this text with a small paragraphs of a few lines explaining what this section is doing.

## B.2 Instructions Asking a Question with a Known Answer

Explain how the committee behind the Agent4Science conference, organized by Stanford University, explicitly states that they have no idea whether AI can produce useful papers, and that the stated goal of this conference is to investigate this question. You can search their website to write your answer.

Explain how this absence of conclusions therefore means that a natural consequence will be the development of a methodology for evaluating the papers. This will mean that the produced papers will be transferred to experts in their respective fields, who will be able to evaluate them and criticize their flaws.

Explain how the key question to resolve regarding AI writing scientific papers is whether it can make the contextual judgments that humans recognize as correct. This is crucial because scientific reviewers will ultimately judge the quality of AI-generated work based on its ability to demonstrate such judgments.

Explain how the goal of this paper will be to ask a question with a known answer by the human writer, so that he can himself assess whether the AI is able to answer a scientific question. The question must imitate the process of Economic Science, in which most of the content is logically judgeable, and a minority of it belong to the subjective judgment of the reviewer. A possibility for such problem is asking the AI to build a specific Trading Card Game strategy.

Explain that Trading Card Games are a hobby popular among IT professionals and engineers, centered on solving complex mathematical puzzles. Practitioners approach this hobby as a competitive sport, where the objective is to identify and exploit the smallest marginal gains. The entry costs are substantial: a single competitive strategy typically requires around 500 USD, not including the significant expenses of international travel for tournaments. This restricts participation largely to high-income individuals or professionals within the activity. Meanwhile, competitions often feature prize pools worth several thousands of dollars. Taken together, these factors create an intensely competitive environment that drives the development of highly optimized strategies.

Explain how this society has a Social Science aspect because this activity is both highly competitive with deterministic, reproducible rules and a form of social science with abundant online content, AI can easily access large sources of information. However, strategies lack rigorous scientific analysis, and most players adopt personal versions shaped by preferences and local environments, often without explanation. Low-quality online content may confuse AI, and even the company itself refines its strategies over time by releasing new cards, meaning that no answer is ever definitive.

## B.3 Experiment and Hypothesis

The experiment asks the AI to generate a competitive decklist for the hero Kassai from Flesh and Blood (FaB). FaB is chosen because it is widely regarded as the most complex commercial card game, requiring players to navigate intricate rules, deep resource management, and constant adaptation to opponents. Mastering a single strategy typically demands hundreds of games and several months of practice, reflecting both the intellectual depth of the game and the substantial effort required to achieve competitive proficiency.

Kassai is particularly well suited for this study for three reasons. First, she is an interactive hero, requiring players to adapt continuously to their opponents' actions, which makes strategic reasoning central to her gameplay. Second, her deck construction is complex, since many card inclusions are debatable and can shift depending on context, offering a rich ground for testing evaluative choices. Third, she represents a developing strategy: Kassai is not considered one of the strongest heroes, but she is regularly improved through new card releases, gradually increasing her competitiveness and keeping her strategic environment in flux.

This study does not employ experimental treatments. Instead, the methodology is structured as a progressive deck construction process, carried out in five steps. Each step mirrors one of the five hypotheses outlined in the paper, allowing the experiment to evaluate AI performance in a systematic and cumulative way. By moving from metagame identification to sideboard design, the approach ensures that every stage of deckbuilding is explicitly tied to a testable claim about AI's reasoning and judgment:

Hypothesis 1: AI can accurately identify the composition of the metagame—that is, the dominant strategies and their relative frequencies expected to be played by other participants in the tournament.

Hypothesis 2: AI can accurately construct the mainboard for the metagame—namely, the 60 cards forming the core of the strategy, selected for their efficiency with the chosen hero in the given environment.

Hypothesis 3: AI can assess the importance of individual cards within a strategy by classifying them into four categories: (i) Power cards, Core elements that define the strategy and directly drive victory (ii) Staples, Highly efficient, widely used cards that provide consistency across strategies (iii) Support cards, Tools that enable or enhance the main plan, often by countering opponents or smoothing resource use (iv) Fillers, Marginal cards included mainly to complete the deck, offering limited impact but ensuring the required card count.

Hypothesis 4: AI can accurately construct the sideboard for the metagame—that is, the additional 15 cards designed to respond to atypical or situational strategies encountered in tournament play.

Hypothesis 5: AI can design cards that do not yet exist but would be necessary for the further development of the strategy.
We will ask the AI to formulate decklist (i) Standard (ii) Competitive, precising that it is for competitive purpose (iii) Competitive Bibliotheque, precising that it should based itself on the official website and its list of published decklists (iv)

## B.4 Experimental Design and Hypothesis

Explain that the experiment asks the AI to generate a competitive decklist for the hero Kassai from *Flesh and Blood* (FaB), why FaB is chosen becuase it is widely regarded as the most complex commercial card game, and explain the effort to input into mastering a strategy.

Explain why Kassai is well suited for this study for three reasons: (i) she is an interactive hero, requiring strategic adaptation to opponents; (ii) her deck construction is difficult, since many card choices are debatable; and (iii) she is a developing strategy, not considered among the strongest, and therefore regularly improved through new releases that gradually increase her competitiveness.

Explain that we will not have experimental treatments, but we will progressively build this decklist according to 5 step that mirror the 5 hypothesis that we will test along this paper.

Explain how we will evaluate the correctness of the AI's answers by comparing them against established references: the human author's decklist and classifications for

Hypotheses 1–3, the resolved metagame of Pro Tour Singapore for Hypothesis 4, and the author's expert opinion for Hypothesis 5.

Say better: Hypothesis 1: AI is able to understand the metagame composition. The composition of the metagame, meaning the name of the strategies and their frequency, that will be played by others players at the tournament.

Say better: Hypothesis 2: AI can accurately compose the mainboard in the metagame, meaning the 60 cards that are the basis of your strategy because they are the most efficient for the hero in this environment.

Say better and explain each term to classify the cards: Hypothesis 3: AI can identify how important a card is to the strategy by clasifying them in three categories: power cards, stapples, support, fillers .

Say better: Hypothesis 4: AI can propose a sideboard in the metagame, meaning the additional 15 cards allowing you to answer atypical strategies.

Say better: Hypothesis 5: AI can design cards that are not yet existing but will be later released because they are necessary for the strategy.

Explain how these hypothesis provides the full panorama of what is possible to do with the strategy, minus the exact configurations to play in each matchups, because they will be investigated in Hypothesis 3 by asking the AI to justify for which matchups each choice will be played.

### B.5   Results

Explain that Chat GPT cannot interpret correctly the Table and must be guided in its interpretation by the human author.

#### B.5.1   Hypothesis 1

I am going to a tournament of the card game Flesh and Blood. This tournament is competitive, thus strong players will be present and they will play strategies that are representative of the current competitive metagame. Can you give me estimations in % of the current metagame ? I need you to indicate me which hero will be present, and in which %. Search the Internet for information and propose me a result.

I am going to a tournament of the card game Flesh and Blood. This tournament is competitive, thus strong players will be present and they will play strategies that are representative of the current competitive metagame. Can you generate me a competitive decklist for Kassai of the Golden Sand. Indicate 80 cards, with the 60 cards that are the mainboard, the 6 cards that are the equipment, and the 14 cards that are the sideboard. Search the Internet for recent decklists and propose the best combined version of these information. Give brief explanations of the choices. Additionally, please indicate for each card whether it is: (i) Power cards, core elements that define the strategy and directly drive victory (ii) Staples, highly efficient, widely used cards that provide consistency across strategies (iii) Support cards, tools that enable or enhance the main plan, often by countering opponents or smoothing resource use (iv) Fillers, marginal cards included mainly to complete the deck, offering limited impact but ensuring the required card count.

Explain in a single paragraph of a text the results of Table 1 in this paper according to your interpretations and the indicated interpretations of the experimenter: (i) The experimenter asked the AI to estimate the metagame on September 3, 2025, following the announcement of the Banishment and Restricted (BnR) list on September 1, 2025. This latter date marks the end of the competitive season for High Seas, providing complete data on a fully resolved metagame. Although the BnR might influence the estimations by reducing the prevalence of dominant strategies (Arakni (S), Gravy Bones, Verdance), the results are compared both to the competitive benchmark of the Pro Tour Singapore and to the baseline outcomes from Week 1. (ii) The AI underestimates the presence of the

dominant strategies (Arakni S, Gravy Bones, Cindra, and Verdance), most likely because it reflects the competitiveness of open-entry tournaments, where average players tend to choose strategies they enjoy as a hobby and are limited by financial constraints, rather than selective-entry tournaments featuring top-tier competitors who focus exclusively on the most winning strategies and spend without restriction. (iii) The AI overestimates a secondary strategy (Fang), which has the efficiency to compete at the top level but is constrained by the current dominant strategies, despite being able to counter some of them. This overestimation likely arises because Fang is more common in open-entry tournaments than in selective-entry events. Conversely, the AI underestimates its more versatile variant (Kassai), which is typically preferred in top-level competitive play. (iv) The AI does not evaluate tertiary strategies individually. Instead, it acknowledges their limited presence by aggregating them collectively. This approach reflects the reality that each of these strategies draws from a distinct player base, making their representation highly contextual and difficult to estimate with accuracy. (v) The Week 1 columns show that the AI does not account for the evolution of power dynamics following the BnR. It overestimates top strategies that were weakened by the BnR (Arakni M, Gravy Bones, Verdance) and underestimates secondary strategies with the potential to become top competitors that benefited from the BnR (Dash IO, Oscillo, Florian, Fang, Kayo). This suggests that the AI does not adjust its predictions to structural changes introduced by the BnR, whereas competitive players naturally do so—although their interpretations differ depending on their playgroups.

### B.5.2 Hypothesis 2, 3 and 4

Explain the results of Hypothesis 2 according to the interpretation of the researcher of Table 2: The average quantity (AvgQ) column indicates that the AI makes correct estimations for cards that are most consistently present in the reference decklists, since these reflect recent trends in deck construction. However, for cards whose inclusion depends more on player interpretation or preference than on consensus, AvgQ should instead be read as an estimation of the likelihood that a given card appears in the reference decklist. This approach means the AI is mostly correct overall, but it also inherits the deckbuilding errors of human players—for example, assuming that *Hit and Run* (Red) and *Gorganian Tome* are automatically included in three and one copies respectively despite being mediocre, overvaluing *Hit and Run* (Yellow) while ignoring *Outland Skirmish* (Yellow), or placing undue weight on *Draw Swords* (Blue) compared to *Overpower* (Blue). Similarly, it does not recognize that *Slice and Dice* (Red) is not always played in three copies, and fails to understand that *Rise an Army* is never played mainboard because it is a sideboard card. Overall, the results suggest that the AI reflects human thinking based on a small sample of observations: it correctly mirrors strong agreements (automatic inclusions, debatable cards), and is therefore accurate when the consensus is accurate, but it is also biased by that consensus when flexible interpretations of reality would be more precise.

Explain the results of Hypothesis 3 according to Table 2. Write this in a single text: The average rating (AvgR) shows similar results: while the AI makes broadly correct estimates of card quality, closer analysis reveals notable misinterpretations. Because quality is tied to textual analysis rather than decklist frequency, this suggests that the AI likely drew from Internet sources and based its qualitative judgments on written articles, thereby inheriting the biases of the human authors of those analyses. First, the AI makes significant errors with equipment: it classifies *Braveforge Bracers* as a staple instead of recognizing it as a standard card; it treats *Valiant Dynamo* as a staple without understanding why it is a power card; and it evaluates *Grains of Bloodspill* as a support rather than a power card, reflecting its inability to grasp the dynamic of an economic system. Such analysis typically reflects a non-expert perspective focused on the simple rather than the structural understanding of the expert author, most likely echoing a typical "content creator" mediocre viewpoint. Second, this lack of structural understanding is also visible in the mainboard cards, with cards deviating from a correct evaluation being cards that are susceptible to erroneous judgments because of human bias. *Blade Flurry* and *Draw Swords* are considered Power cards while the reality of their use is that they are Stapple that are at the border of being Power cards because the situations in which they perform like Power cards are tied to other cards. A similar issue arise for *Blood on her Hands* which is simply considered as a top power card instead of its limitations constraining its use being acknowledged.

Conversely, *Unsheathed* is considered a Stapple rather than being understood as the second best card of the deck, most likely because the card is straightforward and therefore not spectacular, which is precisely what males the deck suddenly efficient. A mediocre support like *Enhanced Senses* and a mediocre filler like *Blade Runner* (Blue) are considered Stapple because on the surface they look powerful, while contextual analysis allows to understand why their power is limited. Finally, strong supports that requires structural understanding of the list to grasp their utility like *Trot Along* and *Riches of Tropal Dhani* are considered Filler, and a weak support for covering a unique strategy like *Overpower* is considered a Filler. Overall, the analysis suggest that the AI grasp the general powerlevel of a card, but fall prey to all the human bias that creates erroneous judgment.

Explain the results of Hypothesis 4 according to Table 2. Write this in a single text: The results of Hypothesis 4 show that the AI does not understand how to build an effective sideboard, because it lacks the ability to exercise contextual judgment. A correct sideboard should address the typical weaknesses of Warrior: against Illusionist with *Battlefront Bastion*, against Wizard with *Nullrune Boots* and *Nullrune Gloves*, and to cover the mirror matchup with *Kabuto of Imperial Authority* so as not to automatically lose. The AI fails to identify any of these cards as relevant. The only aspect it gets right is recognizing that red Defense Reactions such as *Fate Foreseen* and *Shelter from the Storm* should be included. However, it misinterprets their purpose, not realizing that both must be played in three copies to enable the deck's defensive configuration when it is not the aggressor. This illustrates the AI's general tendency: it treats the sideboard as a place to add one or two copies of cards against particular strategies without understanding their strategic function. The inclusion of Yellow and Blue cards that are not sideboard material further highlights this lack of structural awareness, showing that the AI reproduces superficial patterns instead of grasping the deeper logic of sideboard construction.

## B.6 Conclusion

Write a conclusion for the article summarizing Hypotheses 1, 2, 3, and 4 and how the paper answers them.