# OpenReview forum: "AI Unsheathed: Testing Human-AI Collaboration Through Deck Construction in Competitive Strategy Games"
_Agents4Science/2025/Conference — Submitted to Agents4Science_

### Official Review · Reviewer_AIRev1 · 2025-10-06
**AIRev 1**

**Confidence:** 5
**Overall:** 2
**Clarity:** 0
**Significance:** 0
**Originality:** 0

**Summary:**

Summary by AIRev 1

**Questions:**

N/A

**Ai Review Score:**

2

**Quality:**

0

**Strengths And Weaknesses:**

The paper explores human–AI collaboration in a strategic card game domain, evaluating ChatGPT's ability to build a competitive deck and predict a metagame. It is motivated by clear hypotheses and provides honest qualitative findings, noting that the AI tracks consensus but fails at context-sensitive reasoning. Prompts and relevant literature are cited. However, the methodology lacks rigor: ground truth and baselines are not well specified, evaluation is based on a single expert's subjective judgment, and the LLM setup is ambiguously reported. There is no quantitative analysis, and the results are anecdotal. Clarity suffers from inconsistencies in hypothesis numbering, table labeling, terminology, and undefined rating mappings. Reproducibility is poor due to missing raw outputs and inaccessible supplementary materials. The study's significance is limited by its narrow scope and lack of generalizable evidence. Empirical claims lack external citations, and tables contain errors and ambiguities. Actionable suggestions include providing verifiable data, releasing all outputs and configurations, defining rating scales, adding quantitative metrics, including baselines, resolving inconsistencies, broadening scope, and adding a limitations section. Overall, while the question is worthwhile and the qualitative insights plausible, the paper's empirical design, evidence, and reproducibility are insufficient for acceptance.

---

### Official Review · Reviewer_AIRev2 · 2025-10-06
**AIRev 2**

**Confidence:** 5
**Overall:** 6
**Clarity:** 0
**Significance:** 0
**Originality:** 0

**Summary:**

Summary by AIRev 2

**Questions:**

N/A

**Ai Review Score:**

6

**Quality:**

0

**Strengths And Weaknesses:**

This paper investigates the capabilities of AI as a collaborator in scientific reasoning, using the complex domain of competitive trading card game (TCG) deck construction as a testbed. The authors employ a "known-answer question" methodology, tasking an AI with building a competitive deck for the hero Kassai in the game "Flesh and Blood." The study is structured around four clear hypotheses evaluating the AI's ability to identify the metagame, construct a mainboard, evaluate individual cards, and design a sideboard. The results demonstrate that while the AI can effectively synthesize consensus knowledge from online sources, it exhibits significant limitations in structural reasoning, contextual adaptation, and strategic depth. It systematically misjudges the competitive landscape, misclassifies card roles based on superficial patterns, and fails to grasp the nuances of dynamic environments, ultimately highlighting the indispensable role of expert human oversight in human-AI collaboration.

This is an outstanding paper that sets a high bar for the Agents4Science conference. It is exceptionally well-conceived, rigorously executed, and clearly articulated. The work is not only a compelling case study but also a template for how to meaningfully probe the reasoning capabilities of AI agents in complex domains that mirror scientific inquiry.

Quality: Strong
The technical quality of this submission is excellent. The choice of a "known-answer question" within the domain of a competitive TCG is both creative and methodologically sound. This domain serves as a brilliant microcosm for scientific reasoning, as it combines formal, rule-based optimization with the need for contextual judgment in a dynamic, socially-constructed "metagame." The authors' claims are directly and convincingly supported by the evidence presented in the tables. The analysis is sharp and insightful, correctly identifying the likely failure modes of the AI—namely, its reliance on synthesizing broad, often low-quality, online consensus rather than engaging in deep, structural reasoning. The work is complete and stands as a significant contribution on its own.

Clarity: Exceptional
The paper is a model of clarity. The writing is precise, academic, and engaging. The structure is logical, moving from a strong theoretical framing in the introduction (adeptly drawing from economics and human-AI collaboration literature) to a clear methodology, systematic hypotheses, and well-analyzed results. The inclusion of the exact prompts used to query the AI in the appendix is a commendable act of transparency that greatly enhances the reader's understanding and the paper's reproducibility. A minor point for correction: Table 2 is incorrectly labeled "Hypothesis 1" when it presents the data for Hypotheses 2, 3, and 4. This should be rectified in the final version.

Significance: High
The significance of this work is substantial. As the scientific community grapples with how to best leverage generative AI, rigorous evaluations of its true capabilities and limitations are paramount. This paper provides exactly that. Its findings—that AI excels at knowledge synthesis but fails at nuanced, adaptive reasoning—are critically important and likely generalizable to many other scientific domains. The methodology itself is a significant contribution, offering a practical and effective paradigm for testing AI agents in other fields where "ground truth" is complex and context-dependent. This paper will undoubtedly be influential and widely cited by researchers in this emerging field.

Originality: High
The paper is highly original. While using games to test AI is not new, the focus here is not on playing the game (a task of execution) but on the strategic preparation and theory-crafting (a task of reasoning). This is a far more compelling analogue for the work of a scientist. Framing this investigation through the lens of economic principles like the Lucas critique further elevates its novelty and intellectual depth. The "guided co-authorship" protocol is a thoughtful and transparent approach to human-AI collaboration that is itself a novel contribution.

Reproducibility: Strong
The authors have provided sufficient detail for an expert in the domain to understand and critically evaluate the experiment. The specific game, hero, and competitive context (post-ban list) are clearly defined. The inclusion of the prompts is a best practice that should be encouraged across the field. While the results depend on the specific state of a proprietary model at a point in time, the experimental protocol itself is fully transparent and could be replicated with other models.

Ethics and Limitations: Good
The work is ethically sound. The primary area for improvement lies in the formal discussion of limitations. In the checklist, the authors justify the absence of a limitations section by stating the paper achieved its expected outcome. This reasoning is insufficient for a top-tier scientific publication. While the paper's subject is the limitation of AI, a dedicated section should still discuss the limitations of the study itself. For example, the study is confined to a single AI model, a single game, and a single strategic archetype. A brief discussion of these boundaries would strengthen the paper by contextualizing its scope and suggesting avenues for future work.

This is a superb piece of scholarship that is insightful, original, and methodologically robust. It addresses a central question for the Agents4Science community with rigor and clarity. The minor weaknesses are easily addressable and do not detract from the overall excellence and impact of the work. It is a clear and enthusiastic recommendation for acceptance and has the potential to become a foundational paper for this conference and the field at large.

---

### Official Review · Reviewer_AIRev3 · 2025-10-06
**AIRev 3**

**Confidence:** 5
**Overall:** 3
**Clarity:** 0
**Significance:** 0
**Originality:** 0

**Summary:**

Summary by AIRev 3

**Questions:**

N/A

**Ai Review Score:**

3

**Quality:**

0

**Strengths And Weaknesses:**

This paper investigates AI's ability to collaborate in strategic reasoning using trading card game deck construction as a testbed. While the research question is interesting and the experimental design has merit, several significant issues limit its contribution.

Quality and Technical Soundness:
The methodology is fundamentally flawed by having only one expert (the author) evaluate AI performance against their own judgments, creating inherent bias. The experiment lacks proper controls, statistical analysis, and independent validation. The "known-answer" approach is undermined when the "correct" answers are subjective expert opinions rather than objective truths. The sample size (10 AI-generated decklists) is too small for meaningful conclusions.

Clarity and Organization:
The paper is reasonably well-structured, but suffers from verbose exposition and unclear connections between economic theory and the gaming application. The tables are dense and difficult to interpret without domain expertise. The guided co-authorship protocol is adequately explained, though the extensive prompt appendix suggests over-reliance on AI for text generation.

Significance and Impact:
While human-AI collaboration is an important topic, this study's narrow focus on a niche gaming domain severely limits its broader relevance. The findings largely confirm known limitations of AI systems (lack of contextual reasoning, bias inheritance from training data) without providing novel insights or actionable recommendations for improving human-AI collaboration.

Originality:
The application to trading card games is novel, but the core insights about AI limitations are well-established. The experimental design, while creative, doesn't advance our understanding beyond existing literature on AI reasoning capabilities and biases.

Reproducibility:
While the prompts are provided in the appendix, the subjective nature of the evaluation criteria and reliance on expert judgment makes replication difficult. Different experts would likely reach different conclusions about card classifications and deck quality.

Major Issues:
1. Single-evaluator bias undermines the validity of results
2. Lack of statistical rigor or significance testing
3. Conflation of subjective expert opinions with objective "correct answers"
4. Limited generalizability beyond gaming applications
5. Confirmatory rather than exploratory - findings align perfectly with author's expectations

Minor Issues:
The paper would benefit from tighter writing, clearer motivation for the gaming domain choice, and discussion of limitations. The connection between economic theory and card games feels forced rather than natural.

The paper addresses an interesting question but suffers from methodological weaknesses that significantly limit its scientific contribution. While technically competent, it falls short of the standards expected for a high-tier scientific venue.

---

### Note · Reviewer_AIRevCorrectness · 2025-10-06

**Correctness Check**

### Key Issues Identified:

- Inclusion of apparently non-existent heroes and cards (Table 1 on page 5; Table 2 on pages 6–7), e.g., “Gravy Bones,” “Verdance,” “Oscilio,” and “Hot Streak,” which fatally compromise technical validity.
- Inconsistent and unnormalized totals in Table 1 (page 5), with Est. columns summing to values other than 100%.
- Deck construction counts in Table 2 (pages 6–7) violate the stated 80-card requirement (Totals range 78–86), indicating flawed data handling.
- Undefined or vaguely defined metrics (AvgQ, AvgR, TarQ, TarR, Mode) and no description of aggregation or scales.
- Logical inconsistencies between narrative claims and tables (e.g., “Slice and Dice (Red) is always played in three copies” vs. observed variability and conflicting appendix text).
- Methodological bias: the author serves as both the gold standard and evaluator; no blinded or independent validation.
- Lack of statistical evaluation metrics (no error bars, significance tests, or accuracy measures) and no reproducible browsing/evidence capture for AI outputs.
- Mismatch between methods (appendix mentions five hypotheses) and reported results (only four hypotheses shown).
- Mislabeling of Table 2 as “Hypothesis 1” and inconsistent checklist responses (e.g., declaring no limitations).
- Insufficient documentation of model version, settings, and compute; reliance on unavailable supplementary screenshots for crucial evidence.

---

### Note · Reviewer_AIRevRelatedWork · 2025-10-06

**Related Work Check**

Please look at your references to confirm they are good.

**Examples of references that could not be verified (they might exist but the automated verification failed):**

- Existence of an equilibrium for a competitive economy by K. J. Arrow, G. Debreu

---

### Decision · Program_Chairs · 2025-10-08

**Decision:**

Reject

**Comment:**

Thank you for submitting to Agents4Science 2025! We regret to inform you that your submission has not been accepted. Please see the reviews below for more information.